# The Reversion of cg05575921 Methylation in Smoking Cessation: A Potential Tool for Incentivizing Healthy Aging

**DOI:** 10.3390/genes11121415

**Published:** 2020-11-27

**Authors:** Robert Philibert, James A. Mills, Jeffrey D. Long, Sue Ellen Salisbury, Alejandro Comellas, Alicia Gerke, Kelsey Dawes, Mark Vander Weg, Eric A. Hoffman

**Affiliations:** 1Department of Psychiatry, University of Iowa, Iowa City, IA 52242, USA; jim-mills@uiowa.edu (J.A.M.); Jeffrey-long@uiowa.edu (J.D.L.); kelsey-dawes@uiowa.edu (K.D.); 2Behavioral Diagnostics LLC, Coralville, IA 52241, USA; 3Department of Biostatistics, University of Iowa, Iowa City, IA 52242, USA; 4Department of Radiology, University of Iowa, Iowa City, IA 52242, USA; sue-salisbury@uiowa.edu (S.E.S.); eric-hoffman@uiowa.edu (E.A.H.); 5Department of Internal Medicine, University of Iowa, Iowa City, IA 52242, USA; alejandro-comellas@uiowa.edu (A.C.); alicia-gerke@uiowa.edu (A.G.); mark-vanderweg@uiowa.edu (M.V.W.); 6Molecular Medicine Program, University of Iowa, Iowa City, IA 52242, USA; 7Center for Access & Delivery Research and Evaluation, Iowa City VA Health Care System, Iowa City, IA 52242, USA

**Keywords:** smoking cessation, contingency management, DNA methylation, AHRR, cg05575921

## Abstract

Smoking is the largest preventable cause of mortality and the largest environmental driver of epigenetic aging. Contingency management-based strategies can be used to treat smoking but require objective methods of verifying quitting status. Prior studies have suggested that cg05575921 methylation reverts as a function of smoking cessation, but that it can be used to verify the success of smoking cessation has not been unequivocally demonstrated. To test whether methylation can be used to verify cessation, we determined monthly cg05575921 levels in a group of 67 self-reported smokers undergoing biochemically monitored contingency management-based smoking cessation therapy, as part of a lung imaging protocol. A total of 20 subjects in this protocol completed three months of cotinine verified smoking cessation. In these 20 quitters, the reversion of cg05575921 methylation was dependent on their initial smoking intensity, with methylation levels in the heaviest smokers reverting to an average of 0.12% per day over the 3-month treatment period. In addition, we found suggestive evidence that some individuals may have embellished their smoking history to gain entry to the study. Given the prominent effect of smoking on longevity, we conclude that DNA methylation may be a useful tool for guiding and incentivizing contingency management-based approaches for smoking cessation.

## 1. Introduction

Over the past several years, scores of investigative teams have used epigenome wide association studies (EWAS) to identify lifestyle factors and disease processes that hinder healthy aging. The disease focused studies have identified methylation predictors for a variety of discrete aging related pathological processes, such as diabetes and Alzheimer’s disease [1,2,3,4]. Conversely, the lifestyle oriented EWAS’s have identified a number of behaviors and environmental factors associated with premature aging [5,6,7,8]. In particular, the largest, most consistent of these lifestyle findings with respect to longevity are those for smoking [9]. 

Since more conventionally conducted epidemiological studies have shown these same effects of smoking on longevity [10], the results of these more laboratory oriented studies are perhaps not surprising. Still, these newer DNA methylation-based approaches have furthered our understanding of aging by putting a more exact, objective quantification of the extent of the effect of smoking on survival. This more exact understanding has been communicated in two discrete manners. The more commonly used method is through the impact of smoking on epigenetic aging indices. Using a variety of epigenetic clocks, at least a dozen groups have shown that categorical smoking is associated with accelerated epigenetic aging or a reduction in life expectancy [11,12,13,14]. Alternatively, using quantitating methylation at cg05575921, a CpG residue in the aryl hydrocarbon receptor repressor (AHRR) that is highly predictive of smoking status and intensity, we have shown a more exacting relationship between smoking intensity and expected mortality [15].

These smoking induced changes in DNA methylation are not permanent. Both genome-wide and locus specific assessments have shown that the smoking associated changes in methylation at least partially revert in response to smoking cessation [16,17,18,19]. However, the time scale of that reversion is uncertain. Using epigenome wide association analysis, three groups have shown broad, yet locus dependent reversion of changes, particularly at cg05575921, in response to cessation. However, the time scales examined in these studies were on the order of years, and none of these three studies used biochemical verification of smoking cessation. In contrast, in a small group of cotinine verified quitters (*n* = 5), we showed a 5% increase in DNA methylation at cg05575921 after just one month of cessation. Since most of the accelerated aging in certain populations, such as the Framingham Heart Study offspring cohort, is secondary to smoking [20], this last finding suggests that it may be possible to determine the success of smoking cessation by measuring either epigenetic aging or cg05575921 methylation. In addition, it suggests the tantalizing possibility of using changes in epigenetic aging or cg05575921 as a positive motivator for increasing the success of smoking cessation.

Finding new ways, such as offering quantification of the potential increase in longevity, to motivate smokers to quit smoking is important for everyone. From a societal point of view, quitting smoking is an extremely prosocial behavior that decreases the health care costs, and increases the work productivity of an individual [21]. In addition, members of society also benefit from a reduction in second hand smoke exposure [22]. From the smoker’s point of view, quitting is also beneficial. Quitting smoking eliminates the cost of cigarettes, improves survival and removes a potentially stigmatizing behavior from their personal repertoire. Despite these clear benefits, smokers attempting to quit often feel isolated and that they are imposing hardships onto others. Hence, finding ways to encourage smokers to quit is critical.

One method to increase the likelihood of smoking cessation is through using contingency management (CM). CM is an evidence-based behavioral strategy for promoting abstinence from substances of abuse, including tobacco. CM is based on principles of operant conditioning and involves the systematic application of behavioral consequences, using positive reinforcement to support treatment engagement, quitting, and long-term abstinence [22,23]. Financial incentives in the form of cash or vouchers commonly serve as reinforcers, although access to social and occupational rewards are also sometimes used. CM has been investigated across a wide range of settings and populations of tobacco users and has been combined with a variety of behavioral and pharmacological treatment strategies [24,25,26,27]. Systematic reviews and meta-analyses support the effectiveness of CM for improving abstinence during treatment for tobacco and other substance use disorders [28,29].

However, there are several challenges to the real-world implementation of CM. First, in order to reduce the risk of fraud, it is important that only true smokers, and not those only pretending to be smokers, be eligible to receive rewards. In a large study published in 2015, Volpp and colleagues found that at least 6% of subjects who enrolled in an incentive-based smoking cessation study had undetectable levels of cotinine at study intake, with another 14% failing to provide samples for testing, suggesting the possibility that a substantial portion of the subjects in their trial were actually non-smokers [30,31]. Indeed, it is not difficult to visualize individuals smoking a few cigarettes or chewing a piece of nicotine gum prior to screening with a carbon monoxide detector or serum cotinine assay in order to fool clinicians into believing that the client is a smoker eligible for monetary reward. Thus, finding foolproof methods of demonstrating smoking cessation is critical for this use of CM.

Currently, two biological methods are used to objectively assess smoking status, cotinine (COT), and exhaled carbon monoxide (CO) [32]. While these measures are an improvement on self-reporting, they also have limitations hindering their clinical utility. Both CO and COT have short half-lives, with the ability to detect smoking only within the past 3–4 h and 48–72 h, respectively [32]. Additionally, COT is a metabolite of nicotine and thus unable to distinguish the source of exposure, creating false-positives when the patient uses nicotine-replacement therapies or is exposed to second hand smoke. The development of a new biomarker that addressed these shortcomings could be useful in CM of smoking cessation. If that new biomarker was also tightly tied to expected longevity, the change in expected survival resulting from cessation could serve as an additional motivator for patients in CM based treatments to quit smoking.

Conceivably, DNA methylation could suffice for this purpose. However, to use methylation as a motivator or monitor of smoking cessation, it also is necessary to develop affordable, scalable methods to measure that DNA methylation. Genome wide methylation arrays are not practical clinical tools for measuring methylation in this context because of their high cost, slow turnaround time and reliance on complex data handling procedures. However, for this specific purpose, we have developed a rapid, reference-free, precise (standard error of the mean 0.7%) digital PCR (dPCR) assay that is capable of quantifying cg05575921 methylation using DNA prepared from saliva or whole blood [15,33]. Using this assay, we have shown that cotinine-verified lifetime non-smokers of both adolescents and adults have an average cg05575921 methylation of approximately 85% [15,34]. We have also shown that as smoking is initiated, a dose dependent demethylation of cg05575921 occurs as a function of increasing cigarette consumption [34]. Still, whether cg05575921 reverts quickly enough to be used as a measure of smoking cessation success and as a possible motivator for smoking cessation is not known.

In this communication, we describe the short-term pattern of smoking cessation associated reversion of cg05575921 methylation in 20 subjects, who had biochemically confirmed cessation of smoking.

## 2. Materials and Methods 

### 2.1. Study Approval

The protocols and procedures used in this study were approved by the University of Iowa Institutional Review Board (IRB201706713). All subjects who participated provided informed written consent.

### 2.2. Study Participants

The participants in this study were recruited as part of an ongoing clinical trial of sildenafil in reducing pulmonary inflammation in those undergoing smoking cessation therapy (National Clinical Trials NCT02682147). In brief, 67 subjects over the age of 18 who reported smoking more than 10 cigarettes per day were recruited using a series of advertisements distributed to patients and staff from the University of Iowa Hospital and Clinics. In order to qualify for the study, subjects had to complete a brief online survey instrument on smoking habits and report the current consumption of at least 10 cigarettes per day and a total lifetime consumption of at least 5 pack years (Appendix A). Fagerstrom test for nicotine dependence (FTND) scores for each subjects were also calculated from this data [35].

The rules of the CM paradigm used to increase the likelihood of smoking cessation for this trial were described as a portion of the consent procedure. In brief, subjects were instructed to quit smoking as soon as possible and were offered $400 if they successfully quit smoking. Successful quitting was defined as serum cotinine determinations of <10 ng/mL at the 1-, 2- and 3-month clinic visits. Subjects who failed to attend a clinic visit were deemed as treatment failures. Importantly, in order to allow the use of serum cotinine assays to determine smoking cessation, and to minimize the possibility of centrally acting agents interfering with the effects of sildenafil on the lungs, subjects were encouraged to quit “cold turkey” and forbidden to use nicotine replacement, varenicline, or bupropion to quit smoking. However, they were provided brief counselling by a research assistant at each study visit and during weekly phone contacts over the first month of the study.

### 2.3. Laboratory Measures

All subjects were phlebotomized at intake and each clinic visit to provide serum and DNA for these studies. The serum cotinine determinations were conducted by the University of Iowa Diagnostic Laboratories under standard CLIA compliant procedures. DNA methylation at cg05575921 was determined, as previously described, by personnel blind to subject status [15,34]. In brief, whole blood DNA from each subject at each time point was prepared using cold protein precipitation [36]. Then, 1 μg of each sample was bisulfite-converted using a Fast 96 Epitect Kit (Qiagen, Hilden, Germany) and eluted using 70 μL of 10 mM Tris buffer (pH 8.0) according to the manufacturer’s direction. Next a 3 μL aliquot of the bisulfite converted DNA sample was pre-amped using the Smoke Signature^®^ pre-amplification mixture (Behavioral Diagnostics LLC, Coralville, IA, USA). After a 1:3000 dilution, a 5 μL aliquot, containing approximately 10,000 amplicons, was added to a PCR mixture containing the Smoke Signature^®^ assay (Behavioral Diagnostics) and 2 × droplet digital PCR (ddPCR) master mix from Bio-Rad (Carlsbad, CA, USA), portioned into droplets, then PCR amplified. After amplification was complete, the number of droplets containing amplicons with at least one “C” allele, one “T” allele, or neither allele was then determined using a Bio-Rad QX-200 droplet reader, and the absolute ratio of methylated to total CpG methylation at cg05575921 determined by the observed allele counts to a Poisson distribution.

### 2.4. Statistical Analyses

Comparisons between continuous variables were conducted using bivariate analysis. Specifically, serum cotinine and cg05575921 were compared to each other, and then with lifetime average cigarette consumption, past month daily consumption, and FTND. The analyses of between group differences of continuous clinical variables were conducted using ANOVA, with the significance of the difference between groups determined using the Tukey–Kramer honest significant difference (HSD) [37]. The three groups analyzed were non-quitters, quitters with heavy demethylation, and quitters with light demethylation. The regression reversion of methylation was analyzed, and examined through descriptive methods, using linear mixed effects regression [38]. Descriptively, average changes between consecutive visits were calculated and compared for each group. The linear mixed effects regression model included time (days), group (heavy demethylation vs. light demethylation), and the time × group interaction. We also examined the potential effects of sex and age. A random intercept term was included to account for the dependency due to repeated measures.

## 3. Results

The clinical characteristics of the 67 subjects who participated in the study are given in Table 1. In brief, at the intake visit, the subjects averaged approximately 44 years in age with a slight majority of subjects being male (35 males vs. 32 females). They smoked an average of a pack a day and reported an average lifetime consumption of 28 pack years.

A total of 67, 48, 41, and 40 subjects attended the baseline, first, second, and third monthly clinic visits, respectively. Of the 40 subjects who attended the third clinical visit, 20 had serum cotinine levels (<10 ng/mL) consistent with cessation of smoking at all monthly follow-up visits. The remaining 20 subjects had serum cotinine levels indicative of nicotine consumption (>10 ng/mL) and were categorized as “non-quitters” (NQ) for further analyses.

As the first step of our analyses, we compared the two objective markers of smoking to each other and then to the clinical variables. Serum cotinine levels at intake were significantly correlated with cg05575921 levels (Adjusted R^2^ = 0.21, *p* < 0.0002). Serum cotinine levels were also significantly correlated with lifetime average consumption (Adjusted R^2^ = 0.11, *p* < 0.009), past month daily consumption (Adjusted R^2^ = 0.27, *p* < 0.0001), and FTND (Adjusted R^2^ = 0.11, *p* < 0.007). Cg05575921 methylation was significantly associated with lifetime consumption (Adjusted R^2^ = 0.14, *p* < 0.002), past month self-reported daily consumption (Adjusted R^2^ = 0.18, *p* < 0.0004), and FTND (Adjusted R^2^ = 0.21, *p* < 0.0001).

A histogram plot of the initial cg05575921 methylation values in the 47 NQ subjects (Figure 1) shows a roughly unimodal distribution with an average baseline methylation of approximately 49%. In contrast, a histogram of initial cg05575921 methylation values in the 20 subjects who quit smoking shows a distinct bimodal distribution (Figure 1). To better understand this methylation heterogeneity, we further classified quitters into two groups based on the intake methylation mode of each peak in the bimodal distribution. Quitters with cg05575921 methylation values indicative of moderate to heavy smoking, found in the left peak, were categorized as highly demethylated smokers (HDS), with a maximum intake methylation value of 55% (*n* = 11). Quitters with cg05575921 methylation values indicative of light smoking, found in the right peak, were categorized as lightly demethylated smokers (LDS), with an intake methylation value exceeding 55% (*n* = 9). The average intake methylation values between the HDS and LDS quitter groups were striking, 39% and 79% respectively, despite only a trending difference being seen in the intake serum COT values (255 ng/mL vs. 145 ng/mL, *p* < 0.056). The NQ group had significantly higher levels of COT than the LDS group (270 ng/mL vs. 146 ng/mL, *p* < 0.005).

We next examined whether there were differences between these three groups with respect to self-reported smoking variables using ANOVA. Although both lifetime and past month cigarette consumption, as well as FTND scores, were arithmetically lower in the LDS group, there were no significant differences in these measures between the groups.

In our last set of analyses, we examined the pattern of cg05575921 methylation reversion in each of the three groups. There was no significant change over the 3 month period of time of methylation in those who did not quit smoking (Figure 2). However, both the HDS and LDS groups had notably different cg05575921 reversion curves. The changes in methylation (Δβ) between consecutive visits for the LDS and HDS groups are given in Table 2. The average methylation of the HDS group increased with each consecutive visit, with the smallest Δβ occurring from the baseline to visit 1 (1%) and the largest change occurring between visit 2 and 3 (6%). Similarly, the average methylation of the LDS group increased with each consecutive visit, with the smallest Δβ also occurring between the baseline and visit 1 (0.6%), and the largest Δβ (2%) occurring between both visit 1 and 2 and visit 2 and 3. Additionally, the Δβ over the entire 90-day study in both the group average (5% vs. 11%) and the maximum seen in an individual subject (11% vs. 19%) was considerably less in LDS than the HDS group comparatively. Interestingly, a demethylation response was observed in a few subjects in both groups between baseline and visit 1, and visit 1 and visit 2. However, all 20 subjects showed reversion of methylation between visit 2 and visit 3.

Using linear mixed effects regression, we analyzed the relationship between DNA methylation as a function of time from cessation for both the LDS and HDS groups (Figure 3) across the three-month period that the subjects were followed. Time, group, and their interaction were all associated with changes in cg05575921, while sex and baseline age showed no evidence of association. Figure 3 shows the estimated curves for the HDS group (solid red) and the LDS group (solid blue), along with curves for each individual (HDS subject: dashed red; LDS subject: dashed blue) subject. Those in the HDS group had a larger estimated percentage cg05575921 methylation slope than those in the LDS group (0.12%/day (95% CI: 0.096, 0.15) vs. 0.046%/day (95% CI: 0.018, 0.074); *p* < 0.0003).

## 4. Discussion

The proceeding results demonstrate that smoking cessation is associated with an intensity dependent reversion of DNA methylation at cg05575921. Limitations of the study include the small sample size of patients who quit smoking, and the enrichment of the cohort for those of European ancestry.

In this study, a total of 20 subjects had biochemically verified smoking cessation at three visits over a 90-day period. As noted previously, this is not the first study to show reversion of smoking induced DNA methylation as a function of smoking cessation [15,16,17,18,19]. However, the more homogenous quitting pattern observed in the current set of data allows a better understanding of the initial reversion response as a function of time and initial smoking intensity. Our results clearly demonstrate that the reversion response is related to the intensity of smoking, as indicated by the degree of demethylation of cg05575921 at study intake. We did not observe significant effects of age or gender on the demethylation response. Still, the power to detect those effects in this study was relatively limited and there is a clear need for larger scale studies to more conclusively examine the effects of age, gender, and ethnicity, as well as other potentially confounding clinical or genetic variables, on the methylation reversion process.

In our opinion, whereas the results from the more intensively smoking HDS group are easy to conceptualize, the results from the LDS group deserve more scrutiny. Specifically, we are concerned with the possibility that some of the LDS subjects who enrolled in this study may not have truly been daily smokers. In brief, to date, we have shown that the normal range for cg05575921 using the ddPCR assay is above 80%, with nearly 300 COT-verified, daily adult smokers all having methylation levels below 80% [15,39,40]. Yet in this study, three of the nine LDS subjects had initial methylation levels of greater than 81%, and serum cotinine values of 56, 83, and 181 ng/mL, respectively. Of note, two of these three subjects had COT values that have been previously reported to be drastically below the level expected of an individual smoking at an average rate of 10 cigarettes per day [41]. Reassuringly, the amount of reversion of these “light smokers” was relatively small and did not go above the normal range (see Figure 3). Still, the combination of both high methylation and the low serum cotinine values for at least two of these LDS subjects raises the question as to whether some of the subjects who enrolled in this study were truly the “10 cigarette per day” smokers that they claimed to be at study intake.

These concerns that subjects may misrepresent their smoking status to gain monetary reward are not without precedence. Previously, we conducted an unpublished trial of CM in an Iowa community chemical dependency clinic as part of R44CA213507. Despite the use of exhaled carbon monoxide assessment as part of the pre-screening process for the study, members of the clinic’s staff raised concerns that some subjects may have been initiating smoking in order to qualify for the clinical visit compensation ($100 intake, $30 monthly visit compensation) and the two-part $600 incentive given for successful smoking cessation. Given our experiences, and the prior findings of Halpern and colleagues [30], we believe that in the future, it may be prudent to put measures into place that minimize the likelihood of a subject initiating a short term period of smoking in order to qualify for a monetary reward. One way to do this would be by adjusting the reimbursement pattern to reward cessation in heavier smokers more than in light smokers, as indicated by objective DNA methylation levels.

These findings demonstrate the clinical utility of cg05575921 methylation as a biomarker for smoking status, intensity, and for monitoring the success of cessation therapy. This is important because the short half-life of current biomarkers poses a formidable challenge to the success and implementation of CM therapy. Although these results demonstrate a clear reversion within 90-days of cessation, in both light and heavy smokers, understanding the long-term reversion pattern will be necessary if the use of methylation driven incentives at 6 months and 1 year are to be used in the continued reinforcement of smoking abstinence.

Future studies should examine the effects of combining the effects of financial rewards, both direct and indirect, with the numerically driven prospect of living longer. By itself, informing a patient about the arithmetic change in DNA methylation at cg05575921 would likely have little impact. However, if that difference was expressed as a change in expected mortality, this information could be a strong reinforcer of continued abstinence. Prior health and psychology studies have demonstrated that positive images of potential outcomes can increase the rate of smoking cessation [42]. Indeed, it may well be that giving a patient a personalized, concrete estimate of the effect of his/her cessation on expected longevity, combined with the direct and indirect monetary rewards for quitting smoking, may be a powerful set of incentives for motivating more patients to quit smoking. At a simplistic level, this improvement in survival can be calculated by inserting each patients’ age, gender, and initial cg05575921 methylation into previously published Cox regression formulas that use Framingham health study data for predicting the effects of full smoking cessation on survival [9]. However, since beneficial changes in other lifestyle factors, such as decreased alcohol consumption or improved diet, often accompany smoking cessation, and also have strong independent effects on mortality, a more robust estimate of potential increased survival could be achieved by more comprehensive epigenetic assessments of mortality [9]. Alternatively, insurers could replace upfront financial rewards with offers to discount policy premiums to those who quit smoking. Trials to examine and refine the effectiveness and cost–benefit ratio of an integrated survival-economics approach for incentivizing smoking cessation could help formulate a set of strategies for more generalizable clinical implementation. 

Understanding the potential benefits of using a methylation-based approach for smoking cessation is important because of the relatively high cost of determining DNA methylation. Although some commercial laboratories charge over 100 United States dollars for a cotinine assessment, Food and Drug Administration (FDA) cleared urine cotinine “dipsticks” can be readily obtained for less than $1 each, and be performed as a point-of-care (POC) test by trained personnel. Similarly, although CO monitors require frequent calibration, and the use of sterile mouth pieces, CO assessments can also be performed as a POC test by trained personnel. In contrast, DNA methylation assessments will be considerably more costly, and are unlikely to be transformed into a POC testing approach. As a consequence, establishing the cost–benefit ratio of methylation testing as compared to existing approaches will be essential before wider clinical implementation can be considered.

In summary, we demonstrated a rapid reversion of cg05575921 methylation in response to smoking cessation. We suggest that the incorporation of methylation information, both to monitor progress and to incentivize cessation, into CM paradigms for smoking cessation may increase the success rate of treatment and encourage healthy aging.

## Figures and Tables

**Figure 1 genes-11-01415-f001:**
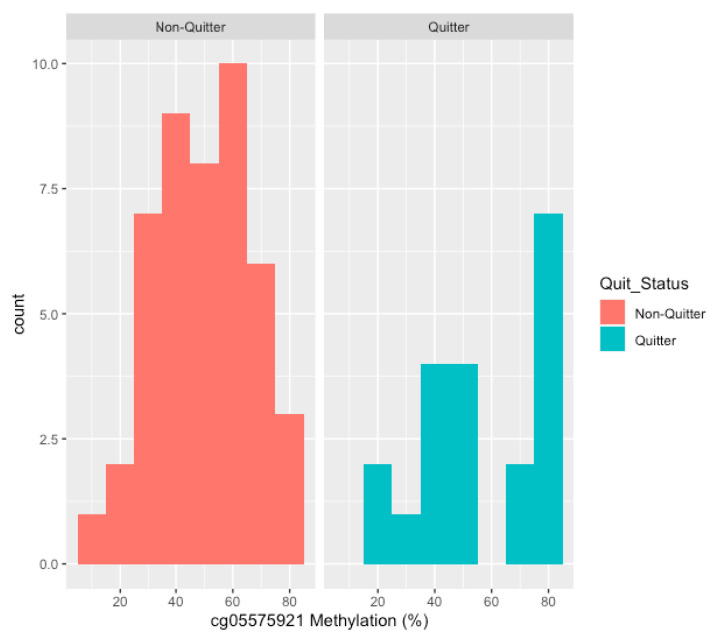
The distribution of the methylation values at study intake. On the left, the distribution of cg05575921 methylation at intake in the non-quitting subjects (*n* = 47). On the right, the distribution of cg05575921 methylation levels at intake in the subjects who quit smoking (*n* = 20).

**Figure 2 genes-11-01415-f002:**
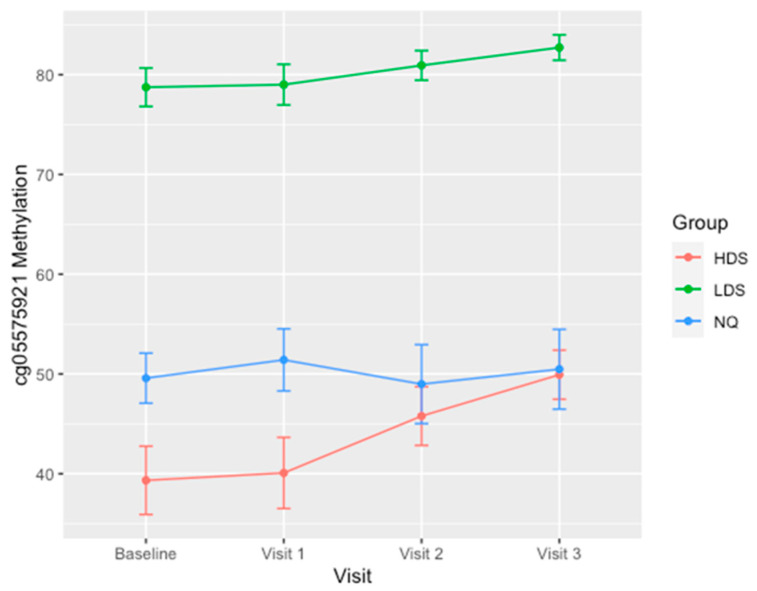
Average methylation values (%) for each group at study intake and each monthly visit. Error bars indicate the standard deviation of methylation for each group. Non-quitter, NQ (*n* = 47), low demethylation smokers (LDS) (*n* = 9) and high demethylation smokers (HDS) (*n* = 11).

**Figure 3 genes-11-01415-f003:**
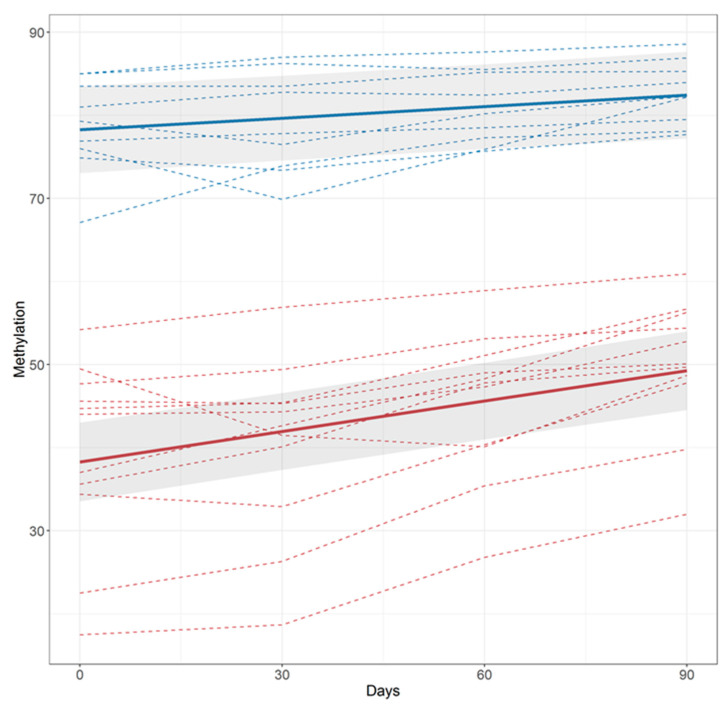
The estimated curves for the high demethylation smokers (HDS) group (*n* = 11, solid red) and the low demethylation smokers (LDS) group (*n* = 9, solid blue) along with curves for each individual (HDS subject: dashed red; LDS subject: dashed blue) as a function of days since quitting. Methylation at cg05595921 is expressed in percent. The rate of reversion (slope) of the HDS group is significantly greater than the LDS group (0.12%/day vs. 0.046%/day; *p* < 0.0003).

**Table 1 genes-11-01415-t001:** Participant Characteristics.

	ALL SUBJECTS	NON-QUITTERS	QUITTERS	
			HDS	LDS
N TOTAL COUNT	67	47	11	9
SEX COUNT				
FEMALE	32	23	3	6
MALE	35	24	8	3
AGE (YEARS)	43.7 ± 10.0	45.4 ± 9.6	40.3 ± 11.1	39.1 ± 8.9
ETHNICITY COUNT				
WHITE	62	43	11	8
HISPANIC WHITE	1	-	-	1
AFRICAN AMERICAN	2	2	-	-
ASIAN	2	2	-	-
PACK YEAR LIFETIME	28.0 ± 18.1	30.6 ± 20.3	25.1 ± 10.3	18.5 ± 7.8
CIGARETTES/DAY LAST MONTH	17.6 ± 8.7	18.4 ± 9.6	18.4 ± 5.8	12.1 ± 2.3
FTND SCORE	3.8 ± 2.2	3.9 ± 2.4	4.2 ±2.2	3.2 ±1.3
INTAKE COT (NG/ML)	251 ± 110	270 ± 112 ^†^	255 ± 78	146 ± 74
INTAKE CG05575921	51.6 ± 18.8	49.3 ± 16.9 ^††^	39.3 ±11.4 ^††^	78.7 ± 5.8

± indicates the standard deviation of the adjacent mean value. All comparisons conducted using ANOVA. ^†^ Different than low demethylation on smokers (LDS) at *p* < 0.05, ^††^ Different than LDS at *p* < 0.01. - is defined as NA.

**Table 2 genes-11-01415-t002:** Changes in methylation between each visit for the high and low demethylation smokers.

	Baseline to Visit 1		Visit 1 to Visit 2		Visit 2 to Visit 3	
	HDS	LDS	HDS	LDS	HDS	LDS
Reversion						
Total number	7	6	8	7	10	9
Mean Δβ	2.13 ± 1.47%	2.55 ± 2.19%	5.38 ± 2.48%	2.63 ± 1.77%	4.36 ± 2.71%	1.79 ± 1.70%
Demethylated						
Total number	3	3	1	2	-	-
Mean Δβ	−3.30 ± 3.38%	−3.47 ± 1.94%	−1.40%	−0.54%	-	-

± indicates the standard deviation of the adjacent mean value. HDS, high demethylation smokers. LDS, low demethylation smokers.

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
