# Peer review of "The Reversion of cg05575921 Methylation in Smoking Cessation: A Potential Tool for Incentivizing Healthy Aging"

_genes, 2020, doi:10.3390/genes11121415_

Round 1

Reviewer 1 Report

Philibert et al propose using methylation levels of cg05575921 (AHRR) to monitor smoking cessation. They have observed increase in the methylation levels (i.e reversion from hypo- to hyper-methylation) of cg05575921 in heavy smokers upon successful cessation, which has already been demonstrated by several earlier studies as referenced by the authors. The novelty of the study lies in the usage of a single CpG site in clinical settings to measure the smoking status, which evidently is more robust and reliable than traditional smoking biomarkers like cotinine and carbon monoxide. From the results, it has been clear that individuals participating in the study for monetary benefits defeat the purpose of the study. In future measuring cg05575921 levels during the recruitment stage might yield better results.   Major changes:
  1. The precision of cg05575921 is pivotal for the success of this prediction strategy. Please expand on the details of line 117. Are you referring to the standard error of the mean or the standard error of the variance?
  2. Please provide more details on the statistical methods used in the "Statistical Analyses" section of the Methods. The current version has an explanation of models in the Results section from 254 to 260.
  3. From the practical and clinical perspective, can you include the details of this strategy in comparison to carbon monoxide and cotinine? For example in terms of time, cost, availability and expertise involved.
  4. As mentioned, providing changes in methylation percentage might not be immediately obvious to the participants. Can you provide what complementary measures/information would be needed to present the results in a more meaningful to the participants?
  Minor changes:
  1. Figure 1: Y-axis is missing. Combining Figures 1 and 2 might be easy to visualize. Also, figures 1 and 2 can be improved by using ggplot2 like the rest of the figures.
  2. Please check for the inadvertent spaces introduced in the sentences.
  3. Please provide abbreviations for HDS and LDS in Table 1, the abbreviations are used for the first time in line 221.
  4. Please tone down on sentences starting with "What is more".
  5. Can authors comment on why no counselling sessions or support groups were implemented to motivate cessation, in addition to the monetary benefits?

Author Response

Reviewer 1.

Comment:Are you referring to the standard error of the mean or the standard error of the variance?

Response:  Standard error of the mean.  We have clarified this matter in the text.  We would be happy to insert more text on the speed and reference-free (which is important) nature of the assay as well if the Reviewer would like these details. In brief, using a droplet digital implementation of the assay, we can obtain results within 5 hours of receiving a sample.  The reference-free nature of the assay is perhaps more critical.  What most researchers do not realize is that the outputs of arrays are normalized with and between samples.  In so many words, the value for one measurement, is dependent on the measurement of another locus or sample.  Generally, the error that is introduced by this normalization process is inconsequential.  However, it can be substantial, particularly for those loci with average methylation values between 25 and 75%.  We have seen the average non-smoking  values for cg05575921 range from 90% to 82% depending on the study.  In contrast, digital PCR outputs are not normalized and are solely dependent on what is put into the sample well being measured.  The average for the biochemically verified non-smokers is always 85-86%; always.

Comment: “Please provide more details on the statistical methods used in the "Statistical Analyses" section of the Methods”.

Response:  Yes, I can see how that would be preferable. We have moved most of the methodological text from the results to the methods and expanded that description.

Comment:From the practical and clinical perspective, can you include the details of this strategy in comparison to carbon monoxide and cotinine? For example, in terms of time, cost, availability and expertise involved”.

Response:  Gladly!   We have placed additional text in the discussion beginning with line 331.

Comment: “As mentioned, providing changes in methylation percentage might not be immediately obvious to the participants. Can you provide what complementary measures/information would be needed to present the results in a more meaningful to the participants?”.

Response:  Absolutely! We recently published an article in Epigenetics that fully details a digital PCR mortality index and describes the method for calculating this change in expected mortality.  As the Reviewer probably appreciates, there are a lot of assumptions in any modeling of this benefit.  Using hazard ratios, we can easily relate the change of methylation to a change in survival.  However, that still is an approximation based on the Framingham cohort.  Truthfully, I think we need to do a bunch of studies to see how this approach can be most effectively employed.  It is odd; but patients always want to “see a number.”  The expanded text covering this subject now covers lines 311-330.

Minor Changes:  We combined Figure 1 and Figure 2, checked and corrected inadvertent spaces (two spaces after each sentence), explicated the two abbreviations in the text,  fixed a typo in the title, toned down the sentences beginning with “what is more” (lines 109 and 121). 

Finally, with respect to the lack of counselling or support question, we note that the clinical center is a referral center and many of our patients are loath to drive back for support groups.  However, they were given brief supportive counseling by a study team member at each study visit.  However, there were no formal groups and the supportive counseling was conducted by a team member who was not a credentialed provider.  We have now described this in the methods.

Reviewer 2 Report

The manuscript by Philibert et al. entitled "The Reversion of cg05575921 Methylation in 2 Smoking Cessation: A Potential Tool for 3 Incentivizing Health Aging" explores the use of AHRR methylation as a marker for short-term follow-up of smoking quitting using ddPCR. The author recruited a small subset of patients (n=67), who were monetarily incentivized to quit smoking. Due to an ongoing trial, they were requested to quit "cold turkey" without any pharmacological aids. Eighteen subjects achieved the quitting goal and were classified as highly vs lightly demethylated smokers. The authors use cotinine levels and questionnaires to follow the process.

Comments:

Comment 1: lines 204-234 My main concern for the authors is how did they control for cell heterogeneity in both saliva and blood samples? Although the changes appear in most of the cells, some authors have shown that most of these changes could be attributed to monocytes, and probably some lymphocytes. Did you adjust for the cell heterogeneity in your models? If not could you add that information to your manuscript? Specifically, add the information between the figure 2 and the cell proportions observed in your samples.

Comment 2: lines 239-252: please also add the cell content in your model here. In addition, what is your limit of detection and the CV%. Could these changes be partially explained to variability in your ddPCR?

Comment 3: lines 254-264: please add the cell components to the model.

Author Response

Comment: “My main concern for the authors is how did they control for cell heterogeneity in both saliva and blood samples? Although the changes appear in most of the cells, some authors have shown that most of these changes could be attributed to monocytes, and probably some lymphocytes. Did you adjust for the cell heterogeneity in your models? If not could you add that information to your manuscript?”.

Response:   All of the Reviewer’s concerns focus on issues related to cellular heterogeneity with the Reviewer also failing to note that the technique employed in the paper is methylation sensitive digital PCR.  Cg05575921 methylation in whole blood samples is not significantly affected by cellular heterogeneity (this is not true for saliva DNA).  Having performed numerous genome wide studies of smoking, I can assure you that values at none of the most highly associated loci, with the exception of those at GPR15, are significantly affected by cellular heterogeneity (we have published at least 6 epigenome wide studies of smoking).  Both Zeilinger (2013) and Shenker  (2012) have noted the absence of effects at cg05575921 with Zeilinger specifically stating that “there was no evidence that any of the blood cell types have significantly different methylation levels that would confound an association with smoking”.  Indeed, if one analyzes our publicly available genome wide data set, one finds that the most significant effects are always are loci whose effects are robust to change in cellular composition.  For the less significant findings, this is often not the case.  But these loci are useless for biomarker studies. Cg05575921, the four markers in the alcohol consumption marker set and the three markers being introduced as part of Epi+Gen CHD™ (www.cardiodiagnosticsinc.com Dogan et al., in submission) are all unaffected by cellular heterogeneity issues in whole blood samples.

Eliminating loci with cellular heterogeneity issues is critical to clinical translation.  If heterogeneity is present, the ddPCR assays would not correlate well with the array values.  Furthermore, the need to correct for cellular heterogeneity would put an insurmountable hurdle in any regulated (e.g. FDA) use of these markers-and I can absolutely assure you of this, since we deal with regulators on this matter.  So, if there are any problems with respect to SNPs or cellular heterogeneity, we simply move on from the marker.  Doing anything else is simply not feasible secondary to regulatory concerns from the FDA or from DHHS (i.e. GINA).

As indicated above, when dealing with saliva DNA, cellular heterogeneity is another matter which we have covered in separate publications.  Please see Andersen et al., 2019 or Dawes et al., 2020.

Comment:In addition, what is your limit of detection and the CV%. Could these changes be partially explained to variability in your ddPCR?

Response:   The published standard error of the mean is 0.7% and with the range of cg05575921 methylation in whole blood ranging from 90% to ~12%, and the observed changes secondary to cessation being ~11% in the heavy smoking group.  Hence, the measurement error here is relatively negligible as compared to the observed changes. It is a digital assay whose precision is based on the Poisson distribution.  Limit of detection has no relevant meaning for these type of assays.  Although we do perform that type of analyses for CLIA (essentially diluting the pre-amp solution), we do not perform the test unless we can be reasonably assured that we are sampling >10,000 independent amplicons.  This subject is covered extensively in our 2018 publication describing the technique.

Round 2

Reviewer 2 Report

The manuscript by Philibert et al explores the use of one CpG in the AHRR as a potential marker for smoking cessation.

I believe the authors have not really evaluated my concern. As a reviewer I also have plenty of EWAS showing the association of AHRR, and I agree this association will not disappear even after cell-type adjustment as all the cells are affected. However, the DEGREE of affectation between the different cell-types is not equal (Bauer, 2016, Bergens, 2019), and there is also literature showing that there are some differences when using whole blood vs PBMCs. My problem is why do you show bimodality in your marker, is this a technical defect?, is it because some subjects have neutrophilia? or maybe is it the opposite and some subjects have lymphocytosis? For this you do not need to run a whole microarray, or investigate SNPs, but only use a CBC (which is usually collected in clinical trials as the one you used for this study). If none of those data are available, please add a sentence explaining that there could be a limitation in the study due to this, but that the magnitude is expected to be minimal. Again, if you have not formally tested that, it is impossible for you to support the claim that there are no differences in your methylation distribution that could be associated to cell heterogeneity.

Author Response

Dear Ms. Wu:

Thank you for your recent e-mail containing the comments from Reviewer 1. In response, please find the attached revised manuscript entitled “The Reversion of cg05575921 Methylation in Smoking Cessation: A Potential Tool for Incentivizing Healthy Aging.” Our detailed response that outlines the changes made in the manuscript are outlined below.

Reviewer 2.

Comment: “I believe the authors have not really evaluated my concern. As a reviewer I also have plenty of EWAS showing the association of AHRR, and I agree this association will not disappear even after cell-type adjustment as all the cells are affected. However, the DEGREE of affectation between the different cell-types is not equal (Bauer, 2016, Bergens, 2019), and there is also literature showing that there are some differences when using whole blood vs PBMCs. My problem is why do you show bimodality in your marker, is this a technical defect?, is it because some subjects have neutrophilia? or maybe is it the opposite and some subjects have lymphocytosis? For this you do not need to run a whole microarray, or investigate SNPs, but only use a CBC (which is usually collected in clinical trials as the one you used for this study). If none of those data are available, please add a sentence explaining that there could be a limitation in the study due to this, but that the magnitude is expected to be minimal. Again, if you have not formally tested that, it is impossible for you to support the claim that there are no differences in your methylation distribution that could be associated to cell heterogeneity

Response: First off, let me personally apologize to the Reviewer. Ignoring the comment was not my intent. I will add text to note this limitation at Line 266 and I inserted the Bergens study as a reference at about line 311 (I would have cited it earlier but that would mean renumbering a good chunk of the text. This way, I only have to renumber with respect to Reference 42-less room for error) so that the readers have an article that reviews the potential for cell-based effects on methylation determinations to which they can refer.

As to the science behind the matter, we have examined both mononuclear cell (ficoll purified) cell pellet DNA and whole blood DNA from the same subjects and not noted a difference. Still, this does not fully answer the question as to whether the addition of a CBC with a WBC differential (which I trust a lot more than I trust the array derived cell counts) would help explain either the current observations or contribute to the error. Strictly speaking, one would need to assess a CBC with diff at each time point and then analyze the resulting data to determine if cell count contributes has a significant effect. Since these are otherwise healthy individuals, I believe that it is unlikely that their WBC would show the kind of shifts that would lead to an introduction of truly significant error. But it would be a good experiment to run because as both of us realize, if you get enough WBCs, there should be several subjects with a large enough left shift to examine to help constrain the effect size and you may trust that I will put that idea into a grant that will be submitted to the NIH this February. There should be some effect if you study sick (for example, those with myelodysplastic syndrome) enough subjects.   And these are exactly the type of individuals that need to quit smoking. Thank you.

As to the two populations of quitters, I am afraid that the reason for the dichotomous distribution is that some of the subjects in the LDS group did not smoke as much as they stated they did. We now have the data from over 1000 subjects from commercial studies (the next paper is in submission elsewhere, and many more subjects from academic studies) to help us constrain the expected cg05575921 result for a given level of cigarette smoking. Frankly speaking, it is clear to us that a number of these subjects in the LDS group were not 10 cigarette per day smokers that they purported to be and as we noted around line 296, we have previously had direct experience with individuals trying to “fake” smoking status to qualify for inclusion in prior incentive based trials. These lower levels of smoking are notable because as the Reviewer realizes, there is no safe level of smoking. But if we are going to monetarily reward smoking cessation, we do not want those who quit to revert rapidly resume smoking. But I am very concerned that the lighter smokers will preferentially resume smoking once the incentive has been received and the study is over. So, overall, whereas I am happy that the science is working out in this area, the implementation of the science will need some nuancing from our more psychologically oriented colleagues to make sure the incentive-based approaches get the maximal bang for the buck.

In summary, we have completely addressed the comments of the Reviewer. We are happy to make additional changes, should they be beneficial to the reader.